# Risk factors for pain and functional impairment in people with knee and hip osteoarthritis: a systematic review and meta-analysis

Sandeep Sandhar,[1] Toby O Smith [ID],[2] Kavanbir Toor,[1] Franklyn Howe,[3] Nidhi Sofat [ID] [1]

[1]Institute for Infection and Immunity, University of London St George's, London, UK
[2]Nuffield Department of Orthopaedics and Musculoskeletal Sciences, University of Oxford, Oxford, UK
[3]Molecular and Clinical Sciences Research Institute, University of London St George's, London, UK

**Correspondence to**
Professor Nidhi Sofat;
nsofat@sgul.ac.uk

## ABSTRACT

**Objective** To identify risk factors for pain and functional deterioration in people with knee and hip osteoarthritis (OA) to form the basis of a future 'stratification tool' for OA development or progression.

**Design** Systematic review and meta-analysis.

**Methods** An electronic search of the literature databases, Medline, Embase, CINAHL, and Web of Science (1990–February 2020), was conducted. Studies that identified risk factors for pain and functional deterioration to knee and hip OA were included. Where data and study heterogeneity permitted, meta-analyses presenting mean difference (MD) and ORs with corresponding 95% CIs were undertaken. Where this was not possible, a narrative analysis was undertaken. The Downs & Black tool assessed methodological quality of selected studies before data extraction. Pooled analysis outcomes were assessed and reported using the Grading of Reccomendation, Assessment, Development and Evaluation (GRADE) approach.

**Results** 82 studies (41 810 participants) were included. On meta-analysis: there was moderate quality evidence that knee OA pain was associated with factors including: Kellgren and Lawrence≥2 (MD: 2.04, 95% CI 1.48 to 2.81; p<0.01), increasing age (MD: 1.46, 95% CI 0.26 to 2.66; p=0.02) and whole-organ MRI scoring method (WORMS) knee effusion score ≥1 (OR: 1.35, 95% CI 0.99 to 1.83; p=0.05). On narrative analysis: knee OA pain was associated with factors including WORMS meniscal damage ≥1 (OR: 1.83). Predictors of joint pain in hip OA were large acetabular bone marrow lesions (BML; OR: 5.23), chronic widespread pain (OR: 5.02) and large hip BMLs (OR: 4.43).

**Conclusions** Our study identified risk factors for clinical pain in OA by imaging measures that can assist in predicting and stratifying people with knee/hip OA. A 'stratification tool' combining verified risk factors that we have identified would allow selective stratification based on pain and structural outcomes in OA.

**PROSPERO registration number** CRD42018117643.

## Strengths and limitations of this study

► This study has been reported in accordance with the Preferred Reporting Items for Systematic Reviews and Meta-Analyses reporting checklist.
► Analyses have been undertaken respecting potential sources of known statistical heterogeneity.
► Searches included both published and unpublished sources of literature to reduce the risk of omitting potentially eligible data.
► There was a paucity of available data to permit meta-analyses of risk factors for pain and functional impairment.
► The variability in methods of assessing risk and reporting of frequency of risk characteristics limited analyses.

disability worldwide caused by OA increased from 10.5 million to 17.1 million, an increase of 62.9%.[2] Current OA treatment lacks any disease-modifying treatments with a predominance to manage symptoms rather than modify underlying disease.[3] The clinical symptoms of OA can be assessed using several questionnaires, the most common of which is the Western Ontario and Mcmaster Universities Osteoarthritis Index (WOMAC).[4–6] Although pain is recognised as an important outcome measure in OA, it is not clear what the optimal assessment tools are in OA and how they relate to other risk factors.

OA has various subtypes and since current therapies cannot prevent OA progression, early detection and stratification of those at risk may enable effective presymptomatic interventions.[7 8] Several methods are used to define, diagnose and measure OA progression, including imaging techniques (eg, plain radiography, CT and MRI). Plain radiography provides high contrast and high-resolution images for cortical and trabecular bone, but not for non-ossified structures (eg, synovial

## INTRODUCTION

It has been reported that over 30.8 million US adults suffer from osteoarthritis (OA).[1] Between 1990 and 2010, the years lived with

fluid).[9] The most recognised radiographic measure classifying OA severity is Kellgren and Lawrence (KL) grading which assesses osteophytes, joint space narrowing (JSN), sclerosis and bone deformity.[10 11] However, it has been argued that MRI may be more suitable for imaging arthritic joints, providing a whole organ image of the joint.[12] Whole-organ MRI scoring method (WORMS) is used in MRI for OA assessing damage, providing a detailed analysis of the joint.

Recently, Outcome Measures in Rheumatology-Osteoarthritis Research Society International (OMERACT-OARSI) have published a core domain set for clinical trials in hip and/or knee OA.[13] Six domains were assessed as mandatory in the assessment of OA, including pain, physical function, quality of life, patient's global assessment of the target joint and adverse events including mortality and/or joint structure, depending on the intervention tested. However, there remains a need to identify risk factors for pain and structural damage in OA so that potential interventions can be studied in a timely manner. The purpose of this systematic review was therefore to identify risk factors for pain, worsening function and structural damage that can predict knee/hip OA development and progression. By identifying risk factors for OA pain and structural damage, tools for stratifying specific disease groups could be developed in the future.

## METHODS

This systematic review has been reported in accordance with the Preferred Reporting Items for Systematic Reviews and Meta-Analyses reporting guidelines.

### Search strategy

A systematic search of the literature was undertaken from 1 January 1990 to 1 February 2020 using electronic databases: Medline (Ovid), Embase (Ovid), Medline, Web of Science and CINAHL (EBSCO). An example of the Embase search strategy of included search terms and Boolean operators is presented in online supplementary file 1. Unpublished literature databases including Clinicaltrials.gov, the WHO International Registry of Clinical Trials and OpenGrey were also searched.

### Study identification

Studies were eligible for inclusion if they were a full-text article that satisfied all of the following:
1. One hundred or more participants analysed in the study (to increase power for comparisons).
2. Convincing definition of OA using American College of Rheumatology criteria,[14] based on symptoms of sustained pain and stiffness in the affected joint, radiographic changes including osteophytes, cartilage loss, bone cysts/sclerosis and JSN, with normal inflammatory markers.
3. Abstract/title that must refer to pain and/or structure in relation to OA as a primary disease.
4. Knee or hip OA.

5. Pain and/or function scores.
6. Joint imaged.
7. Minimum 6-month follow-up of pain/function outcome measures.

Non-English studies, letters, conference articles and reviews were excluded.

The titles and abstracts were reviewed by one reviewer (SS). The full text for each paper was assessed for eligibility by one reviewer (SS) and double-checked by a second (TOS). Any disagreements were addressed through discussion and adjudicated by a third reviewer (NS or FH). All studies that satisfied the criteria were included in the review.

### Quality assessment

To assess the risk of bias and the power of the methodology, the Downs & Black (D&B) tool was applied.[15] These tools assessed the following aspects of each study: reporting quality, external validity, internal validity-bias, selection bias and power. The modified D&B tool was used. Accordingly, the 27-item randomised controlled trial (RCT) version was used for RCTs while the 18-item non-RCT version was used for non-RCT designs (online supplementary file 2). Both 18-item and 27-item tools have been demonstrated to be valid and reliable tools to assess RCT and non-RCT papers.[14] Critical appraisal was performed by one reviewer (SS) and verified by a second (KT). Any disagreements were dealt with by discussion and adjudicated through a third reviewer (TOS). In previous literature, D&B score ranges were given corresponding quality: excellent (scored 26–28); good (scored 20–25); fair (scored 15–19); and poor (scored <14).[14] Item 4 on the non-RCT and item 5 from the RCT tool are scored two points; hence, the total scores equate to 19 and 28 points, respectively. The D&B tool was used to exclude poor quality studies with a score 15/28 or lower in RCTs and 10/19 or lower in non-RCTs.

### Data extraction

Data were extracted including: subject demographic data, study design, pain and function outcome measures, imaging used, OA severity scores, change in pain and function outcomes and change in OA severity scores. After all relevant data had been extracted, authors of these papers were approached to try and attain individual patient data related to baseline and change in pain, function and structural scores for each study. No data were received from authors to inform this analysis.

### Outcomes

The primary outcome was to determine the development of pain and functional impairment for those with knee and hip OA. The secondary outcome was to determine which factors are associated with structural changes in knee and hip OA.

### Data analysis

All data were assessed for study heterogeneity through scrutiny of the data extraction tables. These identified

that there was minimum study-based heterogeneity based on: population, study design and interventions-exposure variabilities for given outcomes. Where there was study heterogeneity, a narrative analysis was undertaken. In this instance, the ORs of all predictor variables were tabulated with a range of OR presented. Where there was sufficient data to pool (two or more studies with data available to analyse) and study homogeneity evident, a pooled meta-analysis was deemed appropriate. As interpreted by the Cochrane Collaboration,[16] when $I^2$ was 50% or greater representing high-statistical heterogeneity, a random-effect model meta-analysis was undertaken. When $I^2$ was less than this figure, a fixed effects model approach was adopted. Continuous outcomes were assessed using mean difference (MD) scores of measures for developing severe OA, whereas dichotomous variables were assessed through OR data. All data were presented with 95% CIs and forest plots.

Due to the presentation of the data, there were minimal data to permit meta-analyses. Where there were insufficient data to pool the analysis (data only available from one study), a narrative analysis was undertaken to assess risk factors for the development of increased pain and functional impairment. Planned subgroup analyses included determine whether there was a difference in risk factors based on: (1) anatomical regions (ie, difference between hip OA and knee OA); (2) geographical region. Analyses were undertaken on STATA V.14.0 (Stata Corp) with forest plots constructed using RevMan Review Manager (RevMan; Computer program; V.5.3. Copenhagen: The Nordic Cochrane Centre, The Cochrane Collaboration, 2014.)

## Patient and public involvement

The research team acknowledges the assistance of both the OA tech network and Engineering and Physical Sciences Research Council. The authors also acknowledge receiving assistance from a meeting that enabled a consensus to be met on the eligibility criteria to be used, and this meeting consisted of the following people: Angela Kedgley, Abiola Harrison, Alan Boyde, Alan Silman, Amara Ezeonyeji, Caroline Hing, Cathy Holt, Debbie Rolfe, Enrica Papi, Freija Ter Heegde, Jingsong Wang, John Garcia, Mark Elliott, Mary Sheppard, Natasha Kapella, Richard Rendle, Shafaq Sikandar, Sherif Hosny, Soraia Silva, Soraya Koushesh, Susanna Cooper and Thomas Barrick. No writing assistance was used.

## RESULTS
### Search strategy

The results of the search strategy are presented in figure 1. In total, 11 010 citations were identified. Of these, 141 papers were deemed potentially eligible and screened at full-text level. Of these, 82 met the selected criteria and were included.[17–98]

## Characteristics of included studies

A summary of the included studies is presented as table 1. This consisted of 31 non-RCTs (27 observational cohort studies/four case-control studies) and 51 RCTs.

In total, 45 767 knees were included in the analysis. This consisted of 13 870 men and 23 497 women; 4 studies did not report the gender of their cohorts.[17–20] Thirty-six studies were undertaken in the USA; 30 were undertaken in Europe; 9 were conducted in Australasia and 7 in Asia. Mean age of the cohorts was 61.7 years (SD: 7.56); 36 studies did not report age.[17 21–54] Mean follow-up period was 35.4 months (SD: 33.6). The most common measures of pain were WOMAC pain (n=55; 50%) and Visual Analogue Scale (VAS) Pain (n=21; 19%). The most frequently used measures of function were WOMAC function (n=52; 44%), physical tests (n=16; 14%) and SF-36 (n=10; 9%).

## Methodological quality assessment

The methodological quality of the evidence was moderate (online supplementary file 2; . Based on the results of the D&B non-RCT tool (31 studies; online supplementary file 2), recurrent strengths of the evidence were clear description of the participants recruited (29 studies; 94%), the representative nature that participants were to the population (31 studies; 100%), and variability in data presented for the main outcomes (31 studies; 100%). Furthermore, the main outcome measures were deemed reliable and valid in all studies (31 studies; 100%) with 89% (27 studies; 87%) studies adopting appropriate statistical analyses for their datasets. Recurrent limitations were not clearly reporting the main findings (20 studies; 65%), issues regarding the representation of the cohort from the wider public (18 studies; 58%) and only 6 studies (19%) basing their sample sizes on an *a prior* power calculation.

The results from the D&B RCT checklist (51 studies; online supplementary file 3) similarly reported findings with strength of the evidence around clear reporting of the cohort characteristics (49 studies; 96%) and interventions (50 studies; 98%), adoption of reliable/valid outcome measures (51 studies; 100%) and reported high compliance to study processes (37 studies; 73%). Recurrent weaknesses included recruiting cohorts which may not have been reflective of the wider population (19 studies; 37%), in clinic settings which may not have represented typical clinical practice (21 studies; 41%) and poorly adjusting for potential confounders in analyses (26 studies; 51%).

## Knee OA
### Narrative review

Findings from the narrative analysis found the following were predictors for worsening joint pain: KL3 or 4 in women (OR: 11.3; 95% CI 6.2 to 20.4), a WORMS lateral meniscal cyst (MC) score of 1 (OR: 4.3; 95% CI 1.2 to 15.4), presence of chronic widespread pain (CWP; OR: 3.2; 95% CI 1.9 to 5.3), increase of ≥2 in WORMS BML

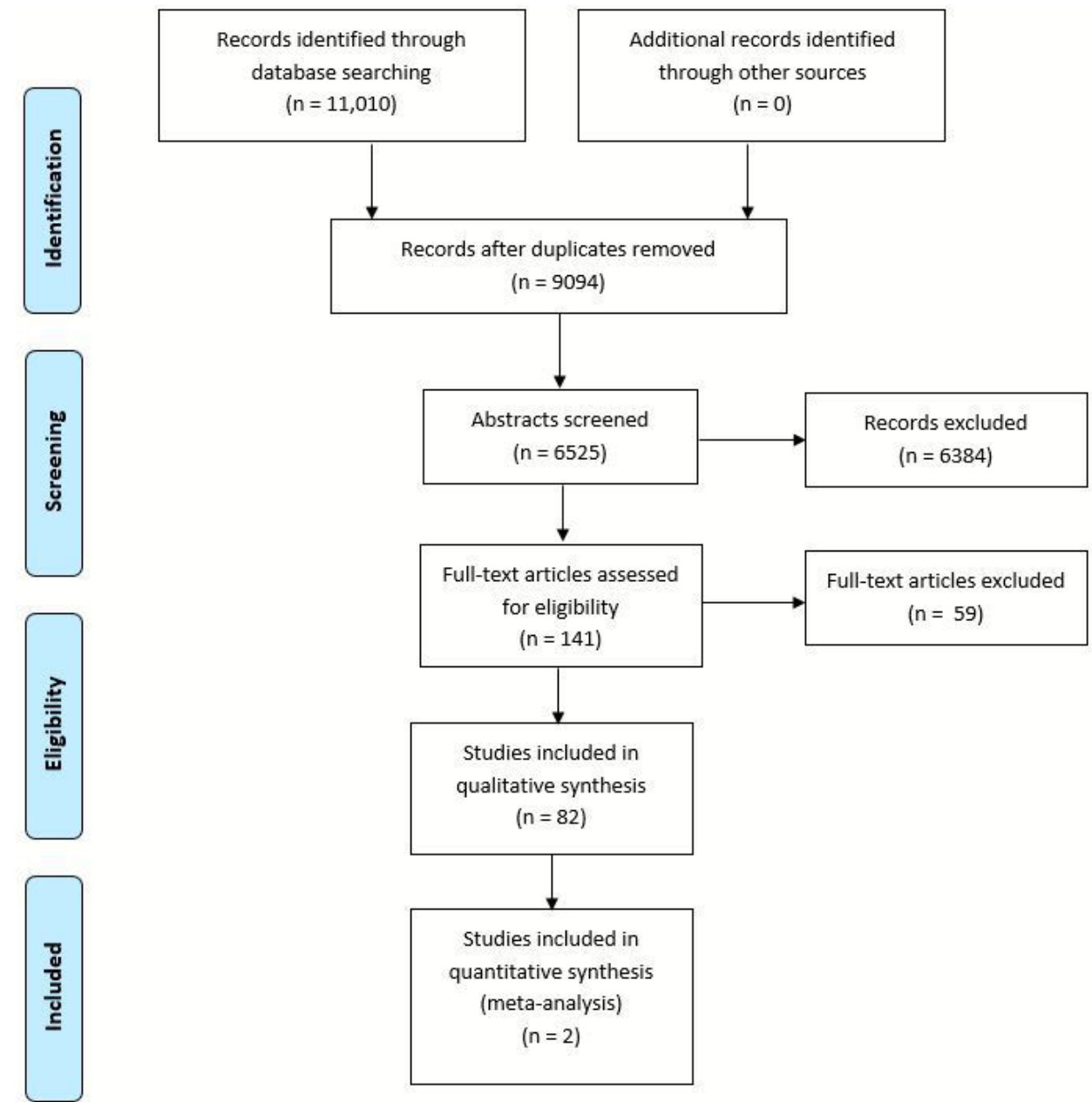

**Figure 1** Preferred Reporting Items for Systematic Reviews and Meta-Analyses flow chart.

score after 15 months (OR: 3.2; 95% CI 1.5 to 6.8), meniscal maceration (OR: 2.8; 95% CI 1.8 to 4.4) or damage ≥2 in WORMS (OR: 1.8; 95% CI 0.9 to 3.6). We also found that the following were the highest predictors of worsening function in people with knee OA: KL of <3 (OR: 3.3; 95% CI 0.7 to 15.9), modified KL 3a (OR: 1.7; 95% CI 0.7 to 3.8), modified KL 4a (OR: 1.5; 95% CI 0.7 to 3.0), presence of osteophytes (OR: 1.3; 95% CI 0.7 to 2.4), female gender (OR: 1.8 (95% CI 1.1 to 3.0) to OR: 2.1 (95% CI 1.2 to 3.5)), ethnicity (OR: 1.03; 95% CI 0.59 to 1.83) and synovitis ≥1 (OR: 1.3; 95% CI 0.8 to 1.9).

### Meta-analysis

Two studies were identified where data could be evaluated for OA risk factors by meta-analysis.[41 67] Three variables significantly associated with the development of knee OA. As illustrated in table 2 and figure 2A–D, age (MD: 1.46, 95% CI 0.26 to 2.66; p=0.02; n=823), KL of ≥2 (MD: 2.04, 95% CI 1.48 to 2.81; p<0.01; n=823) and knee effusion score ≥1 (OR: 1.35, 95% CI 0.99 to 1.83; p=0.05; n=823) were all associated with the development of knee OA based on moderate quality evidence. The variables of gender and BMI were not shown to be significantly associated with the knee OA development (table 2).

Due to the limited availability of data, it was not possible to conduct the planned subgroup analyses to determine whether there was a difference in risk factors based on anatomical or geographical regions.

**Table 1** Characteristics of included studies

| | Study design | Number joints (hip/knees) | Gender (male:female) | Country origin | Mean age (years) | Follow-up duration (months) | Pain outcome measures | Functional outcome measures |
|---|---|---|---|---|---|---|---|---|
| Ahedi et al[54] | Observational cohort | 198 hips | 111:87 | Australia | UTD | 132 | WOMAC Pain | NA |
| Akelman et al[20] | RCT | 107 knee | UTD | USA | 23.5 | 84 | KOOS pain; SF-36 Body pain | SF-36 Physical; AP laxity; IKDC2000 |
| Amin et al[55] | Observational cohort | 265 knees | 152:113 | USA | 67 | 30 | VAS Pain | WOMAC Function |
| Antony et al[56] | Observational cohort | 463 knees | 245:218 | USA | 63 | 24 | WOMAC Pain | NA |
| Arden et al[57] | RCT | 474 knees | 185:289 | UK | 64 | 36 | WOMAC Pain | WOMAC Function |
| Ayral et al[58] | RCT | 665 knees | 259:406 | Australia, Belgium, Canada, Denmark, Finland, France, Hungary, Norway, Spain, UK, USA | 61.3 | 12 | WOMAC Pain | WOMAC Function |
| Baselga Garcia-Escudero and Miguel Hernández Trillos[59] | Observational cohort | 118 knees | 43:75 | Spain | 59.1 | 24 | NRS; WOMAC Pain | WOMAC Function |
| Bevers et al[60] | Observational cohort | 125 knees | 57:68 | The Netherlands | 57 | 24 | WOMAC Pain | WOMAC Function |
| Bingham et al[53] | RCT | 2483 knees | 735:1748 | USA Canada Austria Czech Republic France Germany Hungary Ireland Italy The Netherlands Poland Croatia | UTD | 24 | WOMAC Pain | WOMAC Function |
| Birmingham et al[61] | Observational cohort | 126 knees | 100:26 | Canada | 47.5 | 24 | KOOS Pain | KOOS Function; SF-36 Physical; LEFS |
| Bisicchia et al[52] | RCT | 150 knees | 47:103 | Italy | UTD | 12 | VAS Pain; SF-36 | SF-36 |
| Brandt et al[62] | RCT | 431 knees | 0:431 | USA | 54.9 | 30 | WOMAC Pain; VAS Pain | WOMAC Function |
| Brown et al[51] | RCT | 690 knees | 270:420 | USA | UTD | 32 weeks | WOMAC Pain; NRS weekly pain | WOMAC Function; SF-36 Function |
| Brown et al[50] | RCT | 621 hips | 237:384 | USA | UTD | 32 weeks | WOMAC Pain | WOMAC Function |

Continued

**Table 1** Continued

| | Study design | Number joints (hip/knees) | Gender (male:female) | Country origin | Mean age (years) | Follow-up duration (months) | Pain outcome measures | Functional outcome measures |
|---|---|---|---|---|---|---|---|---|
| Bruyere et al[63] | RCT | 319 knee | 0:319 | Belgium | 64.0 | 36 | WOMAC Pain | WOMAC Function |
| Campbell et al[49] | RCT | 100 knees | 28:72 | Australia | UTD | 120 | American Knee Society Score; WOMAC Pain | American Knee Society Score (function); WOMAC Function |
| Chandrasekaran et al[48] | Case control | 111 hips | 66:45 | USA | UTD | 24 | Modified Harris Hip Score; Nonarthritic hip score; VAS Pin | Modified Harris Hip Score; Nonarthritic hip score; Hip Outcome Score; Sports & ADLs |
| Chandrasekaran et al[47] | Case control | 186 hips | 96:90 | USA | UTD | 24 | Modified Harris Hip Score; Nonarthritic hip score; VAS Pin | Modified Harris Hip Score; Nonarthritic hip score; Hip Outcome Score; Sports & ADLs |
| Conrozier et al[64] | RCT | 205 knees | 88:117 | France | 65 | 26 | WOMAC Pain; NRS walking pain | WOMAC Function |
| Davis et al[19] | Case control | 3132 knees | UTD | USA | UTD | 48 | WOMAC Pain; KOOS Pain | WOMAC Function |
| Dougados et al[46] | RCT | 507 hips | 202:305 | France | UTD | 36 | VAS Pain | Lequesne Index |
| Dowsey et al[65] | Observational cohort | 478 knees | 147:331 | Australia | 70.8 | 24 | IKSS Pain | IKSS Function |
| Eckstein et al[45] | RCT | 1412 knees | 611:801 | Austria | UTD | 48 | WOMAC Pain | NA |
| Ettinger et al[44] | RCT | 439 knees | 131:308 | USA | UTD | 18 | Pain intensity score | Physical Test |
| Felson et al[66] | Observational cohort | 3498 knees | 867:1206 | USA | 61.2 | 30 | WOMAC Pain | PASE |
| Felson et al[67] | Observational cohort | 330 knees | 111:2111 | USA | 62.1 | 15 | NA | Quadriceps strength (N) |
| Filardo et al[43] | RCT | 183 knees | 112:71 | Italy | UTD | 48 | KOOS Pain; IKDC | KOOS Function; Tegner; IKDC |
| Glass et al[42] | Observational cohort | 4648 knees | 918:1486 | USA | UTD | 24 | WOMAC Pain; NRS Pain | WOMAC Function |
| Guermazi et al[41] | Case control | 493 knees | 185:308 | USA | UTD | 60 | WOMAC Pain | PASE |
| Hamilton et al[68] | Observational cohort | 805 knees | 416:289 | UK | 66 | 30 | WOMAC Pain | WOMAC Function |

Continued

**Table 1** Continued

| Study design | Number joints (hip/knees) | Gender (male:female) | Country origin | Mean age (years) | Follow-up duration (months) | Pain outcome measures | Functional outcome measures |
|---|---|---|---|---|---|---|---|
| Hellio le Graverand et al[69] | RCT | 1457 knees | 343:1114 | USA Canada Australia, Belgium, Czech Republic, Germany, Hungary, Italy, Poland, Russian Federation, Slovakia, Spain, Argentina Peru | 61.0 | 180 | Oxford Knee Score | Oxford Knee Score; American Knee Society Score; Tegner |
| Henriksen et al[40] | RCT | 157 knees | 28:129 | Denmark | UTD | 24 | WOMAC Pain | WOMAC Function |
| Hill et al[5] | RCT | 202 knees | 102:100 | Australia | 61 | 12 | KOO Pain | KOOS Function and kinematic assessment |
| Hochberg et al[70] | RCT | 522 knees | 84:438 | France Germany Poland Spain | 62.7 | 24 | WOMAC Pain | WOMAC Function |
| Hoeksma et al[71] | RCT | 109 hips | 33:76 | The Netherlands | 72 | 6 | WOMAC Pain; Huskisson's VAS; EQ-5D Pain | WOMAC Function; EQ-5D Function |
| Housman et al[39] | RCT | 391 knees | 130:261 | USA Canada France UK Germany | UTD | 6 | SF-36 Body Pain; Harris Hip Score; VAS Pain | SF-36 Function; Harris Hip Score; ROM |
| Huang et al[72] | RCT | 264 knees | 39:93 | Taiwan | 62 | 6 | WOMAC Pain | NA |
| Huizinga et al[73] | Observational cohort | 298 knees | 201:97 | The Netherlands | 51 | 12 | VAS Pain | Lequesne index; walking speed |
| Jin et al[6] | RCT | 413 knees | 205:208 | Australia | 63.2 | 24 | WOMAC Pain; VAS Pain | WOMAC Function |
| Kahn et al[74] | Observational cohort | 174 knees | 70:102 | USA | 67.0 | 6 | WOMAC Pain | WOMAC Function |
| Karsdal et al[38] | RCT | 2207 knees | 773:1424 | Denmark | UTD | 24 | WOMAC Pain | WOMAC Function |
| Katz et al[37] | RCT | 330 knees | 143:187 | USA | UTD | 12 | KOO Pain | WOMAC Function; SF-36 Function |
| Kim et al[75] | RCT | 352 knees | 9:153 | Republic of Korea | 68.1 | 144 | WOMAC | Knee Society Knee Score Function; ROM; UCLA Activity |
| Kinds et al[18] | RCT | 565 knees | UTD | The Netherlands | UTD | 60 | WOMAC Pain | WOMAC Function |

Continued

**Table 1** Continued

| Study | Study design | Number joints (hip/knees) | Gender (male:female) | Country origin | Mean age (years) | Follow-up duration (months) | Pain outcome measures | Functional outcome measures |
|---|---|---|---|---|---|---|---|---|
| Kongtharvonskul et al[36] | RCT | 148 knees | 25:123 | Thailand | UTD | 6 | WOMAC Pain; VAS Pain | WOMAC Function |
| Lequesne et al[76] | RCT | 163 hips | 102:61 | France | 63.2 | 24 | VAS Pain | Lequesne Index |
| Lohmander et al[35] | RCT | 170 knees | 52:116 | Bulgaria Canada Croatia Finland Germany Poland Serbia Africa Sweden USA | UTD | 12 | WOMAC Pain | WOMAC Function |
| Maheu et al[8] | RCT | 345 hips | 159:186 | France | 62.2 | 36 | WOMAC Pain; Global Hip Pain | Lequesne Index; WOMAC Function; Global handicap NRS |
| Marsh et al[34] | RCT | 168 knees | 57:112 | Canada | UTD | 24 | WOMAC | WOMAC |
| McAlindion et al[33] | RCT | 146 knees | 57:89 | USA | UTD | 24 | WOMAC Pain | WOMAC Function; Physical Test |
| Messier et al[32] | RCT | 316 knees | 89:227 | USA | UTD | 18 | WOMAC Pain | WOMAC Function; Physical Test |
| Messier et al[77] | RCT | 142 knees | 37:105 | USA | 68.5 | 18 | WOMAC Pain | WOMAC Function; Physical Test |
| Messier et al[78] | RCT | 454 knees | 128:325 | USA | 66 | 18 | WOMAC Pain | WOMAC Function; Physical Test; SF-36 Physical |
| Michel et al[31] | RCT | 300 knees | 146:154 | Switzerland | UTD | 24 | WOMAC Pain | WOMAC Function; Physical Test |
| Muraki et al[79] | Observational cohort | 1558 knees | 553:1005 | Japan | 67.0 | 40 | WOMAC Pain | WOMAC Function; |
| Muraki et al[80] | Observational cohort | 1525 knees | 546:979 | Japan | 67.0 | 40 | WOMAC Pain | WOMAC Function |
| Pavelka et al[30] | RCT | 277 knees; 117 hips | 109:285 | Czech Republic | 58 | 60 | NA | Lequesne Index |
| Pavelka et al[81] | RCT | 202 knees | 45:157 | Czech Republic | UTD | 36 | WOMAC Pain | WOMAC Function; Lequesne Index |
| Pham et al[29] | Observational cohort | 301 knees | 97:204 | France | UTD | 12 | VAS Pain | Lequesne Index |

Continued

**Table 1** Continued

| Study design | Number joints (hip/knees) | Gender (male:female) | Country origin | Mean age (years) | Follow-up duration (months) | Pain outcome measures | Functional outcome measures |
|---|---|---|---|---|---|---|---|
| Podsiadlo et al[28] | Observational cohort | 114 knees | 49:65 | Australia | UTD | 72 | WOMAC Pain | WOMAC Function |
| Rat et al[82] | RCT | 300 knees | 118:182 | France | 67 | 6 | SF-36 Body Pain; OAKHQOL Pain; VAS Pain | Lequense Index; SF-36 Physical; OAKHQOL Physical Activity |
| Raynauld et al[27] | RCT | 123 knees | 44:79 | Canada | UTD | 24 | WOMAC Pain | WOMAC Function |
| Reginster et al[26] | RCT | 212 knees | 50:162 | Belgium | UTD | 36 | WOMAC Pain | WOMAC Function |
| Reginster et al[83] | RCT | 1371 knees | 425:946 | Australia Austria Belgium Canada Czech Republic Denmark Estonia France Germany Italy Lithuania The Netherlands Poland Portugal Romania Russian Federation Spain UK | 62.9 | 36 | WOMAC Pain; VAS Pain | WOMAC Function |
| Riddle and Jiranek[25] | Observational cohort | 467 knees | 209:258 | USA | UTD | 24 | KOOS Pain | WOMAC Function |
| Romagnoli et al[84] | Observational cohort | 105 knees | 16:69 | Italy | 67.7 | 66 | Knee Society Score Clinical; VAS Pain | Knee Society Score Function; ROM |
| Roman-Blas et al[24] | RCT | 158 knees | 26:132 | Spain | UTD | 6 | WOMAC Pain; VAS Pain | WOMAC Function |
| Rozendaal et al[31] | RCT | 222 hips | 68:154 | The Netherlands | UTD | 24 | WOMAC Pain; VAS Pain | WOMAC Function |
| Sanchez-Ramirez et al[85] | Observational cohort | 186 knees | 59:127 | Canada | 61 | 24 | WOAMC Pain | WOMAC Function; Physical Test |
| Sawitzke et al[86] | RCT | 662 knees | 215:447 | USA | 57 | 24 | WOMAC Pain | WOMAC Function |

Continued

**Table 1** Continued

| Study design | Number joints (hip/knees) | Gender (male:female) | Country origin | Mean age (years) | Follow-up duration (months) | Pain outcome measures | Functional outcome measures |
|---|---|---|---|---|---|---|---|
| Skou et al[87] | Observational cohort | 1682 knees | 434:818 | Denmark | 62.2 | 84 | WOMAC Pain | PASE; Physical Test |
| Sowers et al[88] | Observational cohort | 724 knees | 0:363 | USA | 56 | 132 | NA | WOMAC Function; Physical Test |
| Spector et al[89] | RCT | 284 knees | 115:169 | UK | 63.3 | 12 | WOMAC Pain | WOMAC Function |
| Sun et al[90] | RCT | 121 knees | 31:90 | Taiwan | 63 | 6 | WOMAC Pain; VAS Pain | WOMAC Function; Lequesne Index; Physical Test |
| Urish et al[22] | RCT | 336 knees | 96:67 | USA | UTD | 36 | WOMAC | WOMAC |
| Valdes et al[17] | Observational cohort | 860 knees; 928 hips | UTD | UK | UTD | 38 | WOMAC Pain | NA |
| Van der Esch et al[98] | Observational cohort | 402 knees | 64:137 | The Netherlands | 61.2 | 24 | NRS Pain | WOMAC Function; Physical Test |
| Weng et al[91] | RCT | 264 knees | 26:106 | Taiwan | 64 | 12 | VAS Pain | Lequesne Index; ROM; Physical Test |
| White et al[92] | Observational cohort | 2110 knees | 992:118 | USA | 61.0 | 84 | VAS Pain | WOMAC Function |
| Witt et al[93] | RCT | 294 knees | 70:154 | Germany | 64.0 | 12 | WOMAC Pain; SF-36 Body Pain; VAS Pain | WOMAC Function; SF-36 Function |
| Yu et al[21] | Observational cohort | 204 knees | 74:130 | Australia | UTD | 12 | KOOS Pain; VAS Pain | KOOS ADL; Physical Function |
| Yusuf et al[94] | Observational cohort | 74 knees; 31 hips; 11 hip and knees | 19:98 | The Netherlands | 60 | 72 | WOMAC Pain; SF-36 Body Pain; Pain on movement | WOMAC Function; SF-36 Function; Physical Test |

ADLs, activities of daily living; IKDC, International Knee Documentation Committee; KOOS, Knee Injury and Osteoarthritis Outcome Score; LEFS, Lower Extremity Functional Scale; NA, not applicable; NRS, Numerical Rating Scale; OAKHQOL, Osteoarthritis Knee and Hip quality of Life Questionnaire; PASE, Physical Activity Scale for the Elderly; RCT, randomised controlled trial; ROM, range of motion; SF-36, Short Form-36; UTD, unable to determine; VAS, Visual Analogue Scale; WOMAC, Western Ontario and Mcmaster Universities Osteoarthritis Index.

**Table 2** Meta-analysis results: exhibit knee osteoarthritis

| Variable | N | Effect estimate | P value | Statistical heterogeneity (I² %) | GRADE assessment |
|---|---|---|---|---|---|
| Gender | 823 | 0.91 (0.48 to 1.72)* | 0.78 | 87 | Low-quality evidence† |
| Age | 823 | 1.46 (0.26 to 2.66) | 0.02 | 0 | Moderate-quality evidence‡ |
| KL ≥2 | 823 | 2.04 (1.48 to 2.81) | <0.01 | 35 | Moderate-quality evidence‡ |
| Knee effusion score ≥1 | 823 | 1.35 (0.99 to 1.83) | 0.05 | 0 | Moderate-quality evidence‡ |
| BMI | 823 | −0.08 (−0.75 to 0.58) | 0.81 | 0 | Moderate-quality evidence‡ |

*Random effects model analysis.
†GRADE—outcomes downgraded one level due to risk of bias, two level due to imprecision and inconsistency.
‡GRADE—outcomes downgraded one level due to risk of bias.
BMI, body mass index; I2, inconsistency squared; KL, Kellgren Lawrence Scale; N, number of participants in analysis; NE, not estimable.

### Hip OA
#### Narrative analysis
This was based on low-quality evidence. There was no association between the development of hip BML and BMI or age. Predictors for worsening joint pain for people with hip OA included a large acetabular BML

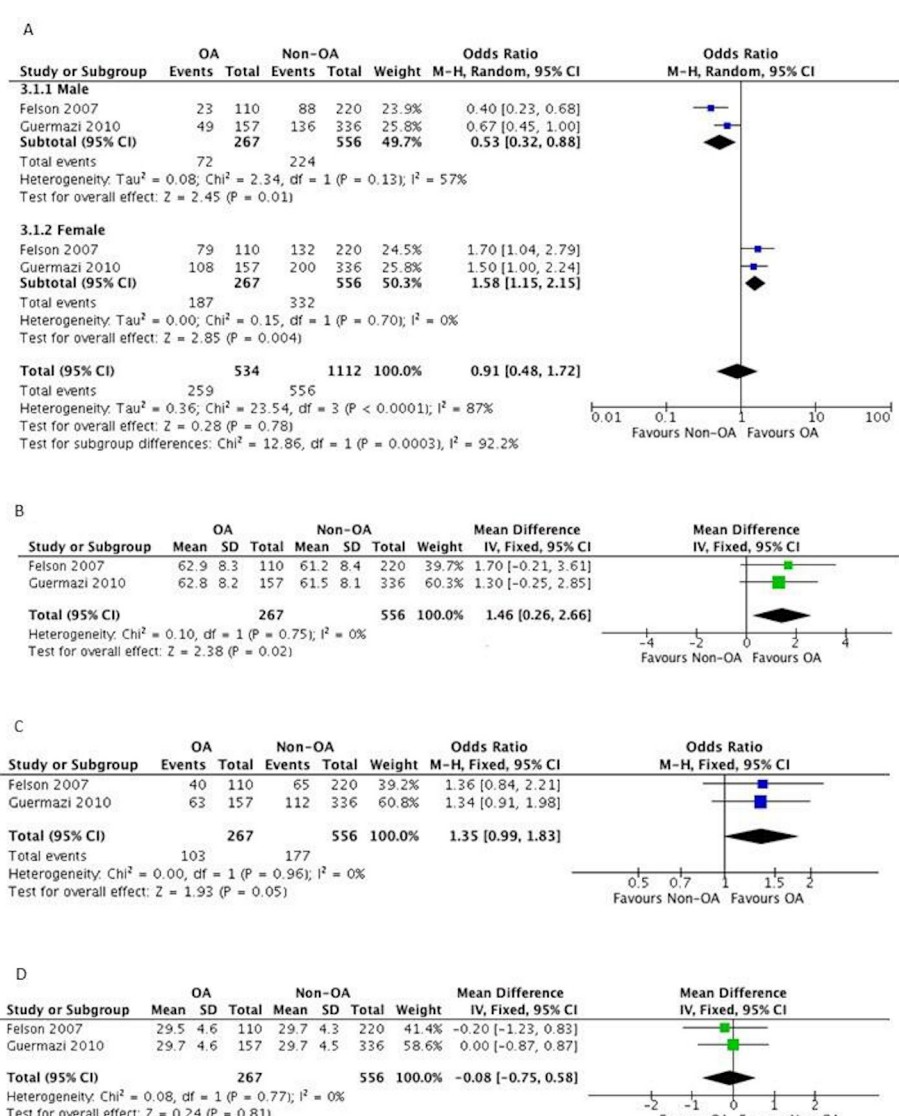

**Figure 2** (A) Forest plot to present the association between gender and presentation of knee osteoarthritis (OA). (B) Forest plot to present the association between age and presentation of knee OA. (C) Forest plot to present the association between knee effusion score greater or equal to 1 and presentation of knee OA. (D) Forest plot to present the association between body mass index and presentation of knee OA.

(OR: 5.2; 95% CI 1.2 to 22.9), a large femoral head BML (OR: 4.4; 95% CI 1.4 to 19.7) with any large hip BML (OR: 4.4; 95% CI 1.5 to 13.2), CWP (OR: 5.0; 95% CI 2.8 to 9.1) and depression (OR: 1.9; 95% CI 1.2 to 2.9). Baseline knee pain score (MD:−1.4; 95% CI −1.6 to −1.2) and baseline hip pain score (MD:−0.7; 95% CI −1.0 to −0.5) were significantly associated with the development of hip BMLs and pain.

### Meta-analysis

There were insufficient data to permit meta-analysis for the hip OA dataset.

## DISCUSSION

Our systematic review and meta-analysis identified risk factors for knee and hip OA pain and structural damage based on evaluation of 82 studies. For the knee, increasing pain in knee OA was associated with KL grade 3 or 4 in women, WORMS lateral MC, presence of CWP, increase of ≥2 in WORMS BML score after 15 months and meniscal maceration. In addition, KL <3, KL 3a, KL 4a, osteophyte presence and female gender were associated with worsening function in people with knee OA. On meta-analysis, age, radiological features (KL score of 2 or more) and knee effusion were associated with development and/or progression of knee OA.

Our meta-analysis identified risk factors that are appreciated only when results were pooled together. These were namely WORMS-defined knee effusion score ≥1. To our knowledge, this is currently the largest and most up to date systematic review of its kind, reviewing 82 primary studies in 41810 participants. Nonetheless, some risk factors from our meta-analysis have been recognised previously. For example, Silverwood *et al* reported previous injuries are associated to developing knee OA, supporting the present analysis.[95] Kingsbury *et al* identified age and KL grade as predictive factors for developing knee OA, supporting the present findings.[96] The meta-analyses provided both novel and supporting findings for risk factors associated with developing and progressing knee OA. A machine learning study assessed risk factors associated with pain and radiological progression in knee OA found that BMLs, osteophytes, medial meniscal extrusion, female gender and urine CTX-II contributed to progression.[97] Nelson *et al's* work is supported by other studies.[95 96] Therefore, the findings of our analysis support previous findings.

After plain radiography, MRI was the most used modality with WORMS as the most common scoring reported for MRI. The MRI Osteoarthritis Knee Score (MOAKS),[99] expanded on WORMS by scoring entire subregions for BMLs rather than each BML, further division of cartilage regions and refined the features assessed in meniscal morphology. Due to this progression from WORMS, having no MOAKS studies included in our final selection was surprising. This could be due to the eligibility criteria being too restrictive. A future systematic review and meta-analysis focusing on the imaging aspect of evaluating OA will be important. In hip OA, the evaluation of BML size and location is essential in predicting pain progression and these can be assessed effectively using MRI. We recommend that all MRI studies for hip OA evaluate BML size and location.

Gait analysis is considered a risk factor for pain/function and was therefore included as a target outcome measure. However, few studies included gait analysis measures, which could not be included in the analysis, perhaps due to the minimum sample size (n=100) being too restrictive.

There were several limitations within our study. First, despite identifying novel risk factors for exhibiting knee OA, a small dataset was pooled together for the meta-analysis (two studies) compared with Silverwood *et al* (34 studies).[93] This was particularly apparent for hip OA where only 12 studies assessed this population.[8 17 23 30 46–48 50 54 71 76 94] Consequently, the small dataset influenced the GRADE assessment that determined the evidence as low to moderate, restricting the strength of the associations of risk factors with OA development and progression. Further work may impact our confidence in the estimated effect, for both studies recruiting participants with hip and knee OA. Second, the eligibility criteria may have been too restrictive, resulting in limited papers including gait analysis or MOAKS. Wet biomarkers were not included in our analyses. Finally, the inability to pool data was partly attributed to variability in methods to report data. Standardising data collection and reporting are important in conducting meta-analyses. We believe the following should be undertaken to improve data pooling in future work: ensuring group comparisons in studies are selected from the same population (people with confirmed OA) to improve internal validity, observational studies should conduct a power analysis to determine sample sizes and all studies should include absolute frequency of events data rather than summary ORs. Such considerations will improve future meta-analyses to identify OA risk factors.

To conclude, our work helps to develop steps towards building a stratification tool for risk factors for knee OA pain and structural damage development. We also highlight the need for collection of core datasets based on defined domains, which has recently also been highlighted by the OMERACT-OARSI core domain set for knee and hip OA.[13] Collection of future datasets based on standardised core outcomes will assist in more robust identification of risk factors for large joint OA.

**Contributors** Conception and design; drafting of the article; critical revision of the article; final approval of the article: NS, FH, TOS and SS. Analysis and interpretation of the data; collection and assembly of data: TOS, SS and KT. Provision of study materials or patients: N/A. Statistical expertise: TOS. Obtaining of funding; administrative, technical, or logistic support: NS, TOS and FH.

**Funding** This study was funded by the Engineering and Physical Sciences Research Council under the reference code 'EP/N027264/1' and The Wellcome Trust ISSF award to NS (Grant number 204809/Z/16/Z).

**Competing interests** None declared.

**Patient consent for publication** Not required.

**Provenance and peer review** Not commissioned; externally peer reviewed.

**Data availability statement** All data relevant to the study are included in the article or uploaded as supplementary information.

**ORCID iDs**
Toby O Smith http://orcid.org/0000-0003-1673-2954
Nidhi Sofat http://orcid.org/0000-0002-6963-6475

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
