## [Reviewer comments · BMJ Open]

ARTICLE DETAILS

TITLE (PROVISIONAL)	Risk factors for pain and functional impairment in people with knee and hip osteoarthritis: a systematic review and meta-analysis
AUTHORS	Sandhar, Sandeep; Smith, Toby O.; Toor, Kavanbir; Howe, Franklyn; Sofat, Nidhi

VERSION 1 – REVIEW

REVIEWER	Le Meur Nolwenn French School of Public Health (EHESP); REPERES research team
REVIEW RETURNED	01-Apr-2020

GENERAL COMMENTS	The authors propose a systematic review and a meta-analysis on risk factors and functional impairment in people with knee and hip osteoarthritis. The study is of interest for those in the field. However, the paper presents several issues that need to be addressed: 1. The object of the research is not always clear:a. It is unclear why the authors write about the predictors for painful hip bone marrow lesion being knee pain and hip pain in their article summary while their aim is to identify pain risk factor for knee and hip osteoarthritis, notably using BML as predictor. It is confusing.b. Why the outcomes section focused only on KOA since the authors also talk about HOA in the title and since the beginning of the article?c. Is there a reason why there are so few results for Hip OA systematic review? This matter should probably be discussed in the discussion2. The abstract is not properly summarizing the methods and the results. In the methods distinction should be made between the narrative and meta-analysis approaches. In the results, the OR presented seem to be all extracted from the global of the >82 studies while most of them are from the meta-analysis of only 2 studies.3. The protocol for the selection of papers to review is well described but the “convincing definition of OA”. Could the authors be more explicit? A reference for the American College of Rheumatology criteria would also be appreciated.4. In the quality assessment of risk bias and the power of the methodology using modified version of Downs and Black’s tool there is also inconsistency. The modified version is said to rely on
--

	a scale of 18 items for observational studies and 27 items for interventional studies but the quality thresholds are 10/19 and 15/28, respectively. What is the difference of 1 item? 5. In the data extraction section, among the authors of the selected papers how many shared their data. Some comment on that subject might be interesting in the discussion. 6. The distinction between the narrative and meta-analysis section is not always clear: a. In the Data analysis section, for the narrative analysis, it is unclear what “all predictor variables were tabulated with a range of OR presented” means as in the result section most of the reported ORs are single values. (Example page 10 Knee OA systematic review and meta-analysis). b. For the meta-analysis approach, what do the authors mean by “sufficient data” to pool? It would be easier for the readers to know in the data analysis section that 2 papers were used for the meta-analysis. c. What is the rationale and formula for the I2 homogeneity index? 7. The presentation of the result also need clarification: a. Are not In the Characteristics of the Included Studies, the authors mentioned 27 observational studies while in the Methodological Quality section they talk about on 37 studies. What is the correct number? The correct number of observational studies should be clearly state at the begin of the Methodological Quality section. b. In the Knee OA systematic review and meta-analysis, where the ORs are coming from? Is there a range or confidence interval? Same question for HOA. c. Figure 2 is barely readable. Minor comments: 1. Line29: “as” should not it be “a”? 2. In the method annotation, the authors should be consistent in the use of abbreviation. In the data analysis section, the used SMD for standardized mean difference while in the text it is MD.
--	---

REVIEWER	Gregory J. Stoddard University of Utah
REVIEW RETURNED	06-Apr-2020

GENERAL COMMENTS	I carefully reviewed your manuscript, as an expert in meta-analysis and biostatistics. Your research expertise and writing was of the highest scholarship. I could not find a single thing that I could recommend to improve it. Your manuscript really does provide the groundwork for future development of a risk stratification tool.
---

VERSION 1 – AUTHOR RESPONSE

Reviewer: 1

Le Meur Nolwenn

Institution and Country

French School of Public Health (EHESP); REPERES research team

Please state any competing interests or state 'None declared':

'None declared'

Please leave your comments for the authors below

The authors propose a systematic review and a meta-analysis on risk factors and functional impairment in people with knee and hip osteoarthritis. The study is of interest for those in the field.

However, the paper presents several issues that need to be addressed:

1. The object of the research is not always clear:

- a. It is unclear why the authors write about the predictors for painful hip bone marrow lesion being knee pain and hip pain in their article summary while their aim is to identify pain risk factor for knee and hip osteoarthritis, notably using BML as predictor. It is confusing.
- b. Why the outcomes section focused only on KOA since the authors also talk about HOA in the title and since the beginning of the article?
- c. Is there a reason why there are so few results for Hip OA systematic review? This matter should probably be discussed in the discussion

2. The abstract is not properly summarizing the methods and the results. In the methods distinction should be made between the narrative and meta-analysis approaches. In the results, the OR presented seem to be all extracted from the global of the >82 studies while most of them are from the meta-analysis of only 2 studies.

3. The protocol for the selection of papers to review is well described but the “convincing definition of OA”. Could the authors be more explicit? A reference for the American College of Rheumatology criteria would also be appreciated.

4. In the quality assessment of risk bias and the power of the methodology using modified version of Downs and Black’s tool there is also inconsistency. The modified version is said to rely on a scale of 18 items for observational studies and 27 items for interventional studies but the quality thresholds are 10/19 and 15/28, respectively. What is the difference of 1 item?

5. In the data extraction section, among the authors of the selected papers how many shared their data. Some comment on that subject might be interesting in the discussion.

6. The distinction between the narrative and meta-analysis section is not always clear:

- a. In the Data analysis section, for the narrative analysis, it is unclear what “all predictor variables were tabulated with a range of OR presented” means as in the result section most of the reported ORs are single values. (Example page 10 Knee OA systematic review and meta-analysis).
- b. For the meta-analysis approach, what do the authors mean by “sufficient data” to pool? It would be easier for the readers to know in the data analysis section that 2 papers were used for the meta-

analysis.

c. What is the rational and formula for the I2 homogeneity index?

7. The presentation of the result also need clarification:

a. Are not In the Characteristics of the Included Studies, the authors mentioned 27 observational studies while in the Methodological Quality section they talk about on 37 studies. What is the correct number? The correct number of observational studies should be clearly state at the begin of the Methodological Quality section.

b. In the Knee OA systematic review and meta-analysis, where the ORs are coming from? Is there a range or confidence interval? Same question for HOA.

c. Figure 2 is barely readable.

Minor comments:

1. Line29: "as" should not it be "a"?

2. In the method annotation, the authors should be consistent in the use of abbreviation. In the data analysis section, the used SMD for standardized mean difference while in the text it is MD.

Reviewer: 2

Reviewer Name

Gregory J. Stoddard

Institution and Country

University of Utah

Please state any competing interests or state 'None declared':

None declared

Please leave your comments for the authors below

I carefully reviewed your manuscript, as an expert in meta-analysis and biostatistics. Your research expertise and writing was of the highest scholarship. I could not find a single thing that I could recommend to improve it. Your manuscript really does provide the groundwork for future development of a risk stratification tool.

VERSION 2 – REVIEW

REVIEWER	Nolwenn Le Meur EHESP
REVIEW RETURNED	25-May-2020

GENERAL COMMENTS	Dear authors, Thank you for having taken into consideration my previous comments. The article has been greatly improved in its form and its content. I, however, have still few comments: 1. In the results' section of the narrative analysis, it would be more informative and more relevant statistically if the ORs were presented along with their confidence interval or p-value.
---

	2. In Table 2, could the authors explain why the heterogeneity index I² was not estimable for some factors? And what was the meta-analysis model used in those cases? The fixed effects model? 3. For the ethnicity variable if white is in parentheses, does that mean that it is the reference? What were the other ethnic group(s) compared to white? 4. It seems that there is a typo in Figure 1 regarding the number of studies included in the quantitative meta-analysis. In Figure 1 the authors wrote n=4 but in the text, Table 2 and Figure 2, I understood that the meta-analysis was performed on 2 studies. 5. Line 1 P7: I believe there is a typo. The maximum score for the 27-item tool should be 28 and not 35. Shouldn't it? 6. PRISMA checklist: references to page and line numbers are requested but section and paragraph numbers are presented by the authors.
--	--

VERSION 2 – AUTHOR RESPONSE

We have addressed, point-by-point, the comments made by the reviewer, which are summarised below:

Comment: 1. In the results' section of the narrative analysis, it would be more informative and more relevant statistically if the ORs were presented along with their confidence interval or p-value.

Response: We have provided 95% confidence interval data for all narrative analysis results (Knee OA; Narrative Review, Page 10, Lines 10-30; Hip OA Narrative Analysis, Page 11, Lines 8-24).

Comment: 2. In Table 2, could the authors explain why the heterogeneity index I² was not estimable for some factors? And what was the meta-analysis model used in those cases? The fixed effects model?

Response: We apologise for this. The factors which were not estimable were not pooled data. These have therefore been removed from the pool outcomes table as this was confusing. These have been corrected incorporated into the narrative analysis (Table 2).

Comment: 3. For the ethnicity variable if white is in parentheses, does that mean that it is the reference? What were the other ethnic group(s) compared to white?

Response: We apologise as this was confusing. This data were based from 1 study and therefore was incorrectly presented in the meta-analysis results. We have therefore removed this (Table 2).

Comment: It seems that there is a typo in Figure 1 regarding the number of studies included in the quantitative meta-analysis. In Figure 1 the authors wrote n=4 but in the text, Table 2 and Figure 2, I understood that the meta-analysis was performed on 2 studies.

Response: Thank you. We have corrected this as recommended (Figure 1).

Comment: 5. Line 1 P7: I believe there is a typo. The maximum score for the 27-item tool should be

28 and not 35. Shouldn't it?

Response: Thank you. We have corrected this as recommended (Methods, Quality Assessment, Page 7, Line 3-6).

Comment: 6. PRISMA checklist: references to page and line numbers are requested but section and paragraph numbers are presented by the authors.

Response: Thank you. We have corrected this and added page and line numbers to all the sections on the updated PRISMA checklist as requested.

We trust that we have addressed the points raised by the reviewer. Should you require any further information, please do not hesitate to contact me.

Professor Nidhi Sofat (on behalf of the authors)

VERSION 3 – REVIEW

REVIEWER	Nolwenn Le Meur EHESP
REVIEW RETURNED	03-Jun-2020
GENERAL COMMENTS	No further comment.